# The Development of a Predictive Model for Postoperative Renal Function in Living Kidney-Transplant Donors

**DOI:** 10.3390/jcm13237090

**Published:** 2024-11-23

**Authors:** Ryo Tanaka, Ayumu Taniguchi, Yoko Higa-Maegawa, Soichi Matsumura, Shota Fukae, Shigeaki Nakazawa, Yoichi Kakuta, Norio Nonomura

**Affiliations:** 1Department of Urology, Osaka University Graduate School of Medicine, 2-2 Yamadaoka Suita, Osaka 565-0871, Japan; rtanaka@uro.med.osaka-u.ac.jp (R.T.); higa@uro.med.osaka-u.ac.jp (Y.H.-M.); matsumura@uro.med.osaka-u.ac.jp (S.M.); fukae@uro.med.osaka-u.ac.jp (S.F.); nakazawa@uro.med.osaka-u.ac.jp (S.N.); nono@uro.med.osaka-u.ac.jp (N.N.); 2Department of Urology, Juntendo University Urayasu Hospital, 2-1-1 Tomioka Urayasu, Chiba 279-0021, Japan; a.taniguchi.kf@juntendo.ac.jp

**Keywords:** renal transplant donors, postoperative renal function, computed tomography volumetry, functional adaptation

## Abstract

**Background/Objectives**: The accurate prediction of postoperative renal function (post-RF) in living kidney donors is essential for optimizing donor safety and long-term health. After nephrectomy, renal function can be significantly altered, owing to the functional adaptation of the remaining kidney; however, the extent of this has not been investigated. This study aimed to examine how various donor factors affect functional adaptation after nephrectomy, and to develop a new predictive model. **Methods**: In total, 310 patients who underwent donor nephrectomy were included. Preoperative split renal function (pre-SRF) of the remaining kidney was measured. Post-RF was measured 1 month after surgery. The functional adaptation rate was calculated from the difference between pre-SRF and post-RF. Multiple regression analysis was performed to develop a predictive formula for post-RF, incorporating donor age and pre-SRF. **Results**: The median age of the donors was 60 years, and 38.7% were men. The median pre-SRF was 36.4 mL/min/1.73 m^2^. The median functional adaptation rate was 26.8%, with donor age, pre-SRF, and a history of hyperuricemia (HUA) being significant predictors of the functional adaptation rate. The equation for post-RF was established as 0.94 × pre-SRF − 0.12 × age + 18.87 mL/min/1.73 m^2^. The estimated post-RF showed a high coefficient of determination (R^2^ = 0.76), with a mean bias of –0.01 mL/min/1.73 m^2^. **Conclusions**: Donor age, pre-SRF, and HUA are key predictors of renal functional adaptation after nephrectomy. The developed formula accurately estimates post-RF, supporting clinical decision-making and donor counseling for living kidney donations.

## 1. Introduction

The increasing prevalence of chronic kidney disease (CKD) and end-stage renal disease (ESRD) presents a significant healthcare challenge worldwide, with a growing need for effective renal replacement therapies. Chronic kidney disease (CKD) affects over 800 million people worldwide, with more than 2 million of them undergoing treatment for end-stage renal disease [1,2]. Kidney transplantation remains the preferred option for ESRD, offering better long-term outcomes and quality of life than dialysis. Living donor kidney transplantation is particularly advantageous due to its high organ quality, well-documented medical history of the donor, and reduced risks of ischemia–reperfusion injury, which contribute to better initial graft function and overall success rates [3,4].

However, although the benefits of living donor transplantation for the recipients are clear, there is growing recognition of the need to evaluate the donor’s remaining kidney function post-donation. Although living kidney donors generally have a prognosis similar to that of the general population, it remains essential to maintain high standards of care to ensure donor safety and long-term health [5]. In contrast, some studies have suggested a potentially increased risk of CKD and ESRD among living donors [6,7,8], as we have also reported [9], necessitating a more nuanced understanding of these risks to ensure informed decision-making and donor follow-up.

In addition, recent studies have emphasized that early postsurgical kidney function significantly influences long-term outcomes and donor prognosis [10,11]. Accurate assessment and prediction of post-donation kidney function are essential to minimize risk and preserve the functionality of the remaining kidney [12]. This approach involves evaluating the donor’s presurgical kidney function to ensure that the remaining kidney has an adequate glomerular filtration rate (GFR) post-surgery. Although models predicting postoperative renal function in donors have been reported previously, these studies are limited by small sample sizes and none have assessed preoperative split renal function. Therefore, we hypothesized that using split renal function evaluated by computed tomography volumetry (CTV) could improve model performance [13,14]. Generally, post-donation kidney function is expected to increase relative to presurgical levels, a phenomenon known as functional adaptation [15]. However, the degree to which various donor factors, such as age, baseline renal function, and comorbidities, affect this adaptation is not well understood.

Therefore, this study aimed to elucidate the influence of specific donor factors on post-donation functional adaptation and develop predictive equations to estimate the donor’s remaining kidney function. Consequently, we hope to contribute to improved donor selection criteria, enhanced preoperative counseling, and better long-term outcomes for living kidney donors.

## 2. Materials and Methods

### 2.1. Study Design and Participants

This was a cross-sectional observational study from a prospective cohort. Participants were adults aged ≥ 20 years who were living kidney donors. We recruited 312 patients who underwent donor kidney harvesting at our institution between March 2008 and March 2022. The exclusion criteria included cases in which computed tomography volumetry could not be performed preoperatively for medical reasons and cases with missing data. In this study, we employed complete case analysis using list-wise deletion to address missing data. The presence of diabetes, hypertension, and hyperuricemia in the donors’ medical history before surgery was assessed based on whether they were prescribed medication. This study was conducted in compliance with the Declaration of Helsinki, Ethical Guidelines for Medical Research Involving Human Subjects, and the Principles of the Declaration of Istanbul, as outlined in the “Declaration of Istanbul on Organ Trafficking and Transplant Tourism”. All facilities were approved by the Central Ethics Review Committee of Osaka University (#21375) on 6 February 2023. Written informed consent was obtained from all the participants.

### 2.2. Evaluation of Preoperative Split Renal Function (Pre-SRF) of Remained Kidney

All patients underwent 1 mm slice contrast-enhanced computed tomography (CT). The renal volume was calculated automatically using volume analyzer software (SYNAPSE VINCENT version 4, FUJI-FILM, Tokyo, Japan).

The estimated GFR (eGFR) was calculated using the equation developed by the Japanese GFR equation based on serum creatinine levels [16].

Pre-SRF was calculated using CT volumetry as follows: [17]

Pre-SRF = (preoperative eGFR) × (residual renal volume)/(total renal volume).

### 2.3. Comparison of Pre- and Postoperative Renal Functions (Post-RF)

The eGFR was calculated from serum creatinine levels measured at 1 week and 1 month postoperatively.

Functional adaptation rate was calculated as follows [15]:

Functional adaptation rate = [(postoperative eGFR) − (pre-SRF)]/(pre-SRF) × 100.

### 2.4. Development and Validation of Equations

We divided the entire cohort 2:1 into the development and validation cohorts to generate a matched cohort set based on gender, age, and renal function. In the development cohort, the least squares regression line for donor age and pre-SRF against post-RF and the coefficient for the combined equation were calculated using multiple regression analysis. We used multiple regression analysis to examine how several factors independently affect the dependent variable. This approach lets us control for overlapping effects, making our results more reliable. The explanatory variables were selected based on the significance of the full model, which included interaction terms. The estimated post-RF was visualized using a Bland–Altman plot.

### 2.5. Statistical Analyses

Data are expressed as means ± standard deviations or as counts and ratios (%). Bee swarm plots were used to analyze the distributions. The normal distribution of each variable was confirmed using qq plots. Continuous variables between the two groups were compared using a two-tailed Student’s *t*-test, and comparisons between multiple groups were performed using one-way analysis of variance with Tukey’s post hoc multiple comparison test. The performance of the prediction formula was evaluated using least squares linear regression by evaluating the slope and regression-line interception. The bias between the predicted and measured post-RF was visualized using Bland–Altman plots. The 95% confidence intervals were calculated by bootstrap resampling with a normal distribution approximation. Statistical significance was defined as *p* < 0.05. Statistical analyses and data visualization were performed using JMP^®^ pro 17.0.

## 3. Results

### 3.1. Characteristics of Participants

The demographic characteristics of the participants are shown in Table 1.

After excluding two participants who lacked CT volumetric data, 310 participants were eligible. In the overall cohort, 190 and 120 were women and men, respectively, with a mean age of 59.3 ± 11.0 years. The preoperative eGFR was 75.2 ± 12.6 mL/min/1.73 m^2^, and 268 (86.5%) of the removed kidneys were left-sided. The preoperative, 1-week postoperative, and 1-month postoperative split renal functions were 37.1 ± 6.6 mL/min/1.73 m^2^, 47.3 ± 8.6 mL/min/1.73 m^2^, and 46.9 ± 8.3 mL/min/1.73 m^2^, respectively.

### 3.2. Preoperative Assessment of Split Renal Function Using Computed Tomography Volumetry

We used CT volumetry (CTV) to evaluate right and left renal volumes and split renal function for the whole cohort.

The CTV showed that the right and left kidney volumes were 143.5 ± 26.1 mL and 152.8 ± 30.5 mL, respectively, with renal volume ratios of 48.5 ± 2.8% and 51.5 ± 2.8% for the right and left kidneys, respectively (Appendix A).

### 3.3. Preoperative Renal Function Correlates with Post-RF

We evaluated the association between pre-SRF and post-RF.

Scatterplot analysis showed that eGFR at 1 week and 1 month after surgery correlated well with pre-SRF (Figure 1a,b).

In contrast, the bias values between the post-RF value at 1 week and 1 month and the pre-SRF value were 10.2 and 9.8 mL/min/1.73 m^2^, respectively (Figure 1c,d, Table 2). In other words, post-RF increased by approximately 10 mL/min/1.73 m^2^ compared to pre-SRF.

### 3.4. Predictors of Change in Pre- and Post-RFs

Post-RF increased compared to pre-SRF in most cases (Figure 2a).

The functional adaptation rate calculated by the above method averaged 26.8% (Figure 2b).

In other words, renal function increased by approximately 25% in patients with a single kidney after surgery from the preoperative assessment of split renal function 1 month after surgery.

Multiple regression analysis of the influence of donor factors on the functional adaptation rate revealed that pre-SRF, donor age, and HUA were significant predictors (Appendix A).

### 3.5. Establishment of Prediction Formula for Post-RF

We established a predictive formula for post-RF using pre-SRF and donor age. Using a development cohort of 208 participants, the post-RF was estimated using multiple regression analyses. Split renal function and donor age were used as predictors. While these two predicters show a slight negative correlation, the correlation is not strong (R^2^ = 0.17). The predictive effects of HUA on post-RF were limited, if anything, because they were not significant in the multiple regression analysis (Appendix A). Therefore, HUA was not included in the prediction of the multiple regression equation for post-RF. Subgroup analysis by donor background for post-RF showed significant correlation with age, pre-SRF, and HUA (*p* < 0.0001, <0.0001, =0.005, respectively). Sex and a history of hypertension and diabetes mellitus also tended to influence post-RF; however, these differences were small (*p* = 0.04, 0.05, 0.08, respectively) (Appendix A).

The combination equation for postoperative renal function was as follows: 0.94 × pre-SRF − 0.12 × age + 18.87.

### 3.6. Performance of Prediction Formula for Post-RF Relative to Measured Post-RF

We analyzed the performance of the established formula in a validation cohort of 102 participants (Figure 3a).

The prediction formula accurately estimated the post-RF, with an R^2^ value of approximately 0.76. The Bland–Altman plots showed that the estimation was performed well throughout the range of post-RF (Figure 3b, Table 3).

## 4. Discussion

In this study, we established a formula to predict post-RF in living kidney donors using pre-SRF and donor age. We demonstrated that post-RF in donors positively correlated with pre-SRF and negatively correlated with donor age. Additionally, we found that HUA tended to result in lower post-RF in donors. However, although statistical analysis indicated some influence of HUA on post-RF, it was minimal; therefore, HUA was not included as a factor in the post-RF multiple regression model. The prediction formula incorporating pre-SRF and donor age demonstrated good predictive ability, with an R^2^ value of 0.76 and low bias.

We also analyzed the “functional adaptation rate”, which we defined as the recovery rate of donor renal function after nephrectomy. The functional adaptation rate was approximately 25%, which is consistent with previous reports [15]. This rate was negatively correlated with pre-SRF and donor age; HUA also had a negative impact.

Previous studies have reported predictive models for donor post-RF using preoperative renal function or CTV [13,14,18]. However, these models often had limitations, such as a correlation coefficient of r = 0.50–0.65 and small sample sizes. When transformed into a coefficient of determination, this value becomes R^2^ = 0.25–0.42. In this study, we included >300 cases and identified significant donor factors affecting post-RF through multiple regression analysis, allowing us to develop a predictive formula with a stronger correlation and lower bias.

Additionally, few studies have examined the impact of donor factors on functional adaptation after nephrectomy. Functional adaptation is believed to result from an increase in glomerular filtration surface area and renal blood flow due to compensatory hypertrophy of the remaining kidney [19,20]. According to one study, the renal plasma flow in the remaining kidney increased by 40% after unilateral nephrectomy [20]. The negative impact of donor age on functional adaptation may be due to physiological changes, such as microvascular changes and glomerulosclerosis associated with aging [21]. Denic et al. demonstrated that aging is a risk factor for underlying abnormalities, such as nephrosclerosis and nephron hypertrophy in donors, and contributes to some functional and structural changes observed [22,23].

Moreover, hyperuricemia is widely recognized as a risk factor for acute kidney injury and CKD [24,25]. Uric acid has also been suggested to cause endothelial dysfunction and increase renal vascular resistance, which may contribute to its effects on functional adaptation.

A previous study reported that early postoperative changes in renal function strongly influenced long-term renal outcomes [26]. Our findings, which demonstrate the influence of pre-SRF, donor age, and HUA on functional adaptation, have significant clinical implications.

In recent years, the use of marginal donors in living donor kidney transplantation has been increasing [27]. The results of this study may serve as a useful reference for donor selection, particularly in older patients or those with HUA.

A limitation of this study is that we focused on early post-RF and did not evaluate changes in renal function over the medium to long term. Additionally, we assessed preoperative diabetes, hypertension, and hyperuricemia based on the presence or absence of medications, so specific numerical values for each condition could not be evaluated. And this study used an eGFR formula rather than measured GFR. We previously reported on this issue and understood its limitations well [28,29]. However, we believe that predicting changes in eGFR will provide valuable information for daily clinical practice. Additionally, the study participants were healthy donors who met the inclusion criteria. Thus, validation in a separate cohort is necessary to determine whether this predictive formula is applicable to patients undergoing nephrectomy for conditions such as renal cancer or pyelonephritis. Moreover, the current study focused on a Japanese population. Although there are no comparative reports for other ethnic groups, previous studies indicate that renal function after donor nephrectomy typically falls to around 60–70% of preoperative levels [30], which is consistent with our findings. We performed internal data partitioning and validation for this study, but further validation with external data is needed to enhance accuracy. Further studies are required in this area.

## 5. Conclusions

Donor age, pre-SRF, and HUA are significant predictors of functional adaptation following living donor kidney donation. Donors who are older or have a history of HUA should be followed up carefully to account for the possibility of poor recovery from postoperative renal dysfunction. Additionally, we have successfully developed a predictive formula for post-RF, which serves as a valuable tool in clinical decision-making and donor counseling. This formula can assist in better assessing the risks and outcomes for potential living donors.

## Figures and Tables

**Figure 1 jcm-13-07090-f001:**
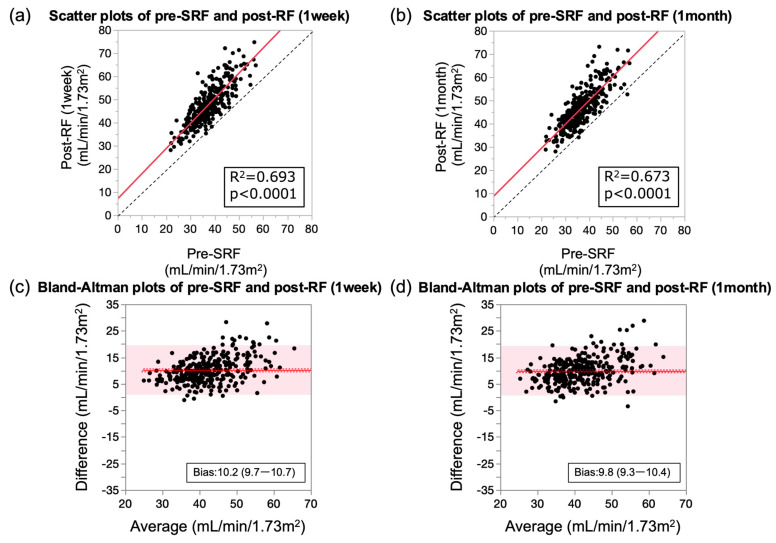
Comparison of association between preoperative split renal function and postoperative renal function at 1 week and 1 month. (**a**,**b**) Scatterplots of preoperative split and postoperative renal functions at (**a**) 1 week and (**b**) 1 month. Solid lines represent regression lines, and black dotted line represents an identical line. R^2^ = coefficient of determination. (**c**,**d**) Bland–Altman plots of preoperative split and postoperative renal functions at (**c**) 1 week and (**d**) 1 month after surgery. Solid red line indicates the mean of difference, red dotted lines represent the 95% confidence interval of the mean of difference, and red shaded areas show the 95% limits of agreement, respectively. Pre-SRF, preoperative split renal function; post-RF, postoperative renal function.

**Figure 2 jcm-13-07090-f002:**
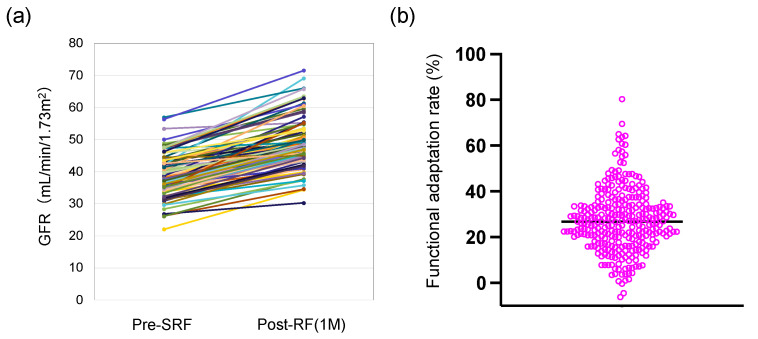
Change in preoperative split and postoperative renal functions 1 month after surgery. (**a**) Changes in preoperative split and postoperative renal functions 1 month after surgery. Lines in different colors represent the changes in renal function for each patient before and after surgery. (**b**) Bee swarm plots of functional adaptation rate. Functional adaptation rate is rate of change in postoperative renal function relative to preoperative split renal function. Pink circles represent the functional adaptation rate for each patient, and black line indicates the mean value. GFR, glomerular filtration rate; pre-SRF, preoperative split kidney function; post-RF, postoperative kidney function.

**Figure 3 jcm-13-07090-f003:**
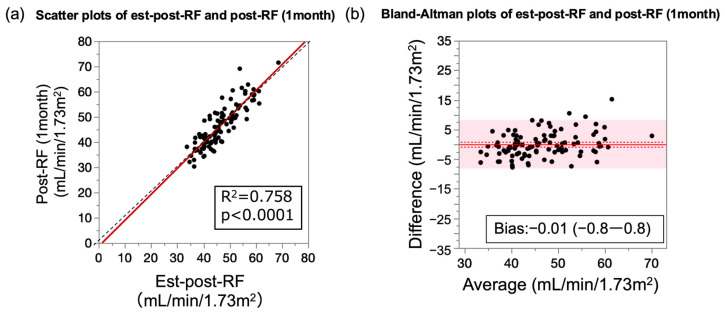
Correlation between predicted and measured postoperative renal function in validation cohort. (**a**) Scatterplots of measured postoperative and predicted postoperative renal functions. Red lines indicate regression lines obtained using least squares method. Black dotted line represents an identical line. R^2^ = coefficient of determination. (**b**) Bland–Altman plots of postoperative and predicted renal functions. Solid red line indicates the mean of difference, red dotted lines represent the 95% confidence interval of the mean of difference, and red shaded areas show the 95% limits of agreement, respectively. Est-post-RF, estimated postoperative renal function.

**Table 1 jcm-13-07090-t001:** Characteristics of participants.

	Total	Development	Validation	*p*-Value
n	310	208	102	
Age, y	59.3 ± 11.0	58.6 ± 11.5	60.7 ± 10.0	0.12
Gender (female/male)	190/120	133/75	57/45	0.17
Body mass index, kg/m^2^	22.8 ± 2.8	22.7 ± 2.8	23.3 ± 3.4	0.26
Body surface area, m^2^	1.62 ± 0.17	1.61 ± 0.16	1.63 ± 0.18	0.36
Diabetes mellitus	8 (2.6)	7 (3.4)	1 (1.0)	0.18
Hypertension	83 (26.8)	53 (25.5)	30 (29.4)	0.46
Hyperuricemia	28 (9.0)	15 (7.2)	13 (12.8)	0.12
Removed kidney side (left/right)	268/42	185/23	83/19	0.08
Serum creatinine level, mg/dL	0.71 ± 0.14	0.70 ± 0.14	0.73 ± 0.15	0.13
eGFR, mL/min/1.73 m^2^	75.2 ± 12.6	75.7 ± 12.4	74.2 ± 13.1	0.31
Pre-SRF, mL/min/1.73 m^2^	37.1 ± 6.6	37.2 ± 6.5	37.0 ± 6.9	0.49
Post-RF (1 week), mL/min/1.73 m^2^	47.3 ± 8.6	47.5 ± 8.4	46.8 ± 9.0	0.47
Post-RF (1 month), mL/min/1.73 m^2^	46.9 ± 8.3	47.2 ± 8.3	46.4 ± 8.4	0.45

Data, n (%) or mean ± SD. eGFR, creatinine-based estimated glomerular filtration rate based on Japanese formula; pre-SRF, preoperative split renal function; post-RF, postoperative renal function. *p*-values were given for difference between development and validation cohorts.

**Table 2 jcm-13-07090-t002:** Correlation and difference parameters of preoperative split and postoperative renal functions.

Equation	R^2^	*p*-Value	Slope (95% CI)	Bias (95% CI)
Post-RF (1 week)	0.69	<0.0001	1.09 (1.01–1.17)	10.2 (9.7–10.7)
Post-RF (1 month)	0.67	<0.0001	1.03 (0.95–1.11)	9.8 (9.3–10.4)

R^2^, coefficient of determination; CI, confidence interval. Slopes were determined using least squares regression analysis. Bias was calculated as mean difference between preoperative split and postoperative renal functions. Post-RF, postoperative renal function.

**Table 3 jcm-13-07090-t003:** Correlation and difference parameters of estimated and measured postoperative renal functions.

Equation	R^2^	*p*-Value	Slope (95% CI)	Bias (95% CI)	RMSE (95% CI)
Est-post-RF	0.76	<0.0001	1.03 (0.92–1.14)	−0.01 (−0.83–0.82)	4.15 (3.61–4.91)

R^2^, coefficient of determination; CI, confidence interval; RMSE, root-mean-square error. The slopes were determined using the least squares regression analysis. Bias was calculated as the mean difference between the estimated and measured postoperative renal functions. Est-post-RF, estimated postoperative renal function.

## Data Availability

Anonymized or aggregated data will be shared upon legitimate requests from academic researchers for research purposes, depending on the nature of the request, merit of the proposed research, and intended use. The steering committee reviews the proposed use for approval.

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
