# Peer review of "The Development of a Predictive Model for Postoperative Renal Function in Living Kidney-Transplant Donors"

_jcm, 2024, doi:10.3390/jcm13237090_

Round 1
Reviewer 1 Report
Comments and Suggestions for Authors
Authors present quite interesting study about the potential role of novel predictive model for postoperative renal function in living kidney donors. This is of special interest, especially for potential donors and caregivers to help to predict kidney status after surgery.
Comments:
1) why post-RF was not analyzed/confirmed in radiological test, as before the surgery?
2) if you say about the 'hyperuricemia', what is the normal value in your lab? please modify that; similarly, what was the value of BP to diagnose hypertension?
3) L41L: I suggest to change into 'Living donor kidney transplantation...';
4) Table 1: please use full name of 'diabetes mellitus', or add 'serum creatinine level';
5) please do not repeat full names when abbreviation was already explained (L182),
6) please be more specific about data presentation and clearly state which reached statistical significance, otherwise please try to omit them (L191-193).
Author Response
Authors present quite interesting study about the potential role of novel predictive model for postoperative renal function in living kidney donors. This is of special interest, especially for potential donors and caregivers to help to predict kidney status after surgery.
Comments:
- why post-RF was not analyzed/confirmed in radiological test, as before the surgery?
→ Thank you for your comment. As stated in the manuscript, we evaluated preoperative kidney volume ratio using CTV, combining this with preoperative eGFR to assess pre-SRF. Since the patient has a single kidney postoperatively, it is not typical to evaluate renal function using CTV; instead, we assessed post-RF using eGFR.
- if you say about the 'hyperuricemia', what is the normal value in your lab? please modify that; similarly, what was the value of BP to diagnose hypertension?
→ For the preoperative history of hyperuricemia and hypertension, we used the presence of medication as an indicator. As such, we did not include individual patient values for uric acid or blood pressure in this study, which we have now acknowledged as a limitation by adding it to the Methods and Discussion sections. (P2L87-89, P8L317-319)
- L41L: I suggest to change into 'Living donor kidney transplantation...';
→ Thank you for your suggestion. We have revised this section. (P2L47)
- Table 1: please use full name of 'diabetes mellitus', or add 'serum creatinine level';
→ Thank you for your suggestion. We have made this change. (Table 1)
- please do not repeat full names when abbreviation was already explained (L182),
→ Thank you for your comment. We have updated this accordingly. (P6L227)
6) please be more specific about data presentation and clearly state which reached statistical significance, otherwise please try to omit them (L191-193).
→ Thank you for your comment. We have added specific p-values. (P6L238, 240)
Reviewer 2 Report
Comments and Suggestions for Authors
Introduction requires more (specific) data
Methodological Biases exist
Some Refs are missing
(The Authors must see my remarks)

Author Response
Introduction requires more (specific) data
→ Thank you for the suggestion. We have added specific data and references in the Introduction. (P1L39-P2L46)
Methodological Biases exist
→ We have added this point to the Discussion as a limitation. (P8L326-331)
Some Refs are missing
(The Authors must see my remarks)
→ Thank you for pointing this out. We have added the missing references in the indicated sections. (P2L62, P3L115, P3L120)
We have also addressed the other minor revisions suggested. (P1L18, P3L111, P3L125, P3L136-137, Table 1, P4L178, Figure 1 Legend)
Reviewer 3 Report
Comments and Suggestions for Authors
Although renal transplantation is the most suitable replacement therapy for patients with chronic kidney disease, there remains significant scope for research, particularly to improve survival rates for both recipients and living donors.
This study seeks to investigate the impact of various donor-related factors on renal functional adaptation. Based on the analyses conducted, the study aims to develop a predictive model for postoperative renal function. A prospective analysis of 310 patients was carried out, concluding that donor age, preoperative renal function, and the presence of hyperuricemia are significant predictors of functional adaptation rates.
The study appropriately presents both background information as well as its methods and results. The conclusions drawn are consistent with the findings obtained. In its current form, the study is a strong candidate for publication. Below are some minor considerations:
- Does the information provided in Table S1 pertain to all 310 cases, or only to the development cohort?
- It would be valuable to discuss the implications of the Bland-Altman plot results. Does the observed bias indicate an increase in postoperative RF compared to preoperative SRF?
- In line 182, the reference should be to Table S2, not S1.
Author Response
Although renal transplantation is the most suitable replacement therapy for patients with chronic kidney disease, there remains significant scope for research, particularly to improve survival rates for both recipients and living donors.
This study seeks to investigate the impact of various donor-related factors on renal functional adaptation. Based on the analyses conducted, the study aims to develop a predictive model for postoperative renal function. A prospective analysis of 310 patients was carried out, concluding that donor age, preoperative renal function, and the presence of hyperuricemia are significant predictors of functional adaptation rates.
The study appropriately presents both background information as well as its methods and results. The conclusions drawn are consistent with the findings obtained. In its current form, the study is a strong candidate for publication. Below are some minor considerations:
- Does the information provided in Table S1 pertain to all 310 cases, or only to the development cohort?
→ Table S1 contains data for all 310 cases. We have clarified this. (P4L172)
- It would be valuable to discuss the implications of the Bland-Altman plot results. Does the observed bias indicate an increase in postoperative RF compared to preoperative SRF?
→ You are correct. We have added a clear explanation of the results. (P5L198-199)
- In line 182, the reference should be to Table S2, not S1.
→ Thank you for catching this error. We have corrected it. (P6L228)
Reviewer 4 Report
Comments and Suggestions for Authors
Here are several potential improvements for the study and manuscript titled “Development of a Predictive Model for Postoperative Renal Function in Living Kidney Transplant Donors”:
- Clarity in Abstract and Introduction:
- Give additional details on the limitations of current predictive models would help readers understand the study's novelty.
- Rephrase the objective in simpler terms.
- Methodology Details:
- it would be beneficial to explain why each statistical method (e.g., multiple regression) was chosen for specific analyses.
- Include a section on handling missing data, outliers, or data transformations .
- Study Limitations:
- The discussion does not fully address the potential limitations related to the specific demographics of the donor cohort (e.g., Japanese population). Expand on how results might differ in other populations.
- Add a dedicated section discussing assumptions made in the predictive model (e.g., linearity between variables.
-Expanded Discussion on Results:
- Add insights into the biological reasons behind the findings—like how age impacts kidney adaptability—for example.
- Include a comparison of the model's performance (e.g., R² value) against similar studies or predictive formulas for helping contextualize the model's effectiveness.
- Validation and Model Robustness:
- The study uses a development and validation cohort, but an external dataset validation (e.g., another institution's donor data) could solidify the model's robustness.
- I suggest the need for future prospective studies to test the model’s performance over a longer postoperative period
- Visual Clarity in Figures:
o Add more explanatory labels or legends in scatterplots and Bland-Altman plots.
Author Response
Here are several potential improvements for the study and manuscript titled “Development of a Predictive Model for Postoperative Renal Function in Living Kidney Transplant Donors”:
- Clarity in Abstract and Introduction:
- Give additional details on the limitations of current predictive models would help readers understand the study's novelty.
→ Thank you for your suggestion. We have added this information in the Introduction. (P2L64-68)
- Rephrase the objective in simpler terms.
→ Thank you for the suggestion. We have revised it accordingly. (P1L15-17)
- Methodology Details:
- it would be beneficial to explain why each statistical method (e.g., multiple regression) was chosen for specific analyses.
→ We have added this to the Methods section. (P3L128-130)
- Include a section on handling missing data, outliers, or data transformations .
→ We have added this information in the Methods section, as well as in the Results. (P2L86-87, P4L163-164)
- Study Limitations:
- The discussion does not fully address the potential limitations related to the specific demographics of the donor cohort (e.g., Japanese population). Expand on how results might differ in other populations.
→ Thank you for your comment. We have added this to the Limitations. (P8L326-331)
- Add a dedicated section discussing assumptions made in the predictive model (e.g., linearity between variables.
→ We have added this in the Results section. (P6L233-235)
-Expanded Discussion on Results:
- Add insights into the biological reasons behind the findings—like how age impacts kidney adaptability—for example.
→ We have added this in the Discussion. (P8L301-304)
- Include a comparison of the model's performance (e.g., R² value) against similar studies or predictive formulas for helping contextualize the model's effectiveness.
→ Thank you. We have added this in the Discussion. (P7L288-289)
- Validation and Model Robustness:
- The study uses a development and validation cohort, but an external dataset validation (e.g., another institution's donor data) could solidify the model's robustness.
→ We appreciate this suggestion and have added it to the Limitations. (P8L329-330)
- I suggest the need for future prospective studies to test the model’s performance over a longer postoperative period
→ We have included this as a limitation and view it as a future direction. (P9L316-317)
- Visual Clarity in Figures:
Add more explanatory labels or legends in scatterplots and Bland-Altman plots.
→ We have revised Figures 1 and 3 accordingly.